# Crystal Structures of Putative Flavin Dependent Monooxygenase from *Alicyclobacillus Acidocaldarius*

**Hyunjin Moon [1], Sungwook Shin [1,2] and Jungwoo Choe [1,*]**

[1]   Department of Life Science, University of Seoul, Seoul 02504, Korea;
      noel1130@hotmail.com (H.M.); swshin330@naver.com (S.S.)
[2]   Repigen Inc. R&D center, 17 Techno 4-ro, Yuseong-gu, Daejeon 34013, Korea
[*]   Correspondence: jchoe@uos.ac.kr; Tel.: +82-2-6490-5627; Fax: +82-2-6490-2664

**Abstract:** Flavin dependent monooxygenases catalyze various reactions to play a key role in biological processes, such as catabolism, detoxification, and biosynthesis. Group D flavin dependent monooxygenases are enzymes with an Acyl-CoA dehydrogenase (ACAD) fold and use Flavin adenine dinucleotide (FAD) or Flavin mononucleotide (FMN) as a cofactor. In this research, crystal structures of *Alicyclobacillus acidocaldarius* protein formerly annotated as an ACAD were determined in Apo and FAD bound state. Although our structure showed high structural similarity to other ACADs, close comparison of substrate binding pocket and phylogenetic analysis showed that this protein is more closely related to other bacterial group D flavin dependent monooxygenases, such as DszC (sulfoxidase) and DnmZ and Kijd3 (nitrososynthases).

**Keywords:** flavin dependent monooxygenase; Acyl-CoA dehydrogenase; FAD

## 1. Introduction

Flavin-dependent monooxygenases are group of enzymes that catalyze insertion of oxygen using molecular oxygen and are involved in a wide range of biological processes [1,2]. They catalyze various reactions, such as hydroxylation, Baeyer–Villiger oxidation, sulfoxidation, epoxidation, and halogenation [3,4]. In order to perform these reactions, flavin dependent monooxygenases generate a reactive intermediate by forming a covalent bond between molecular oxygen and the flavin [5].

Flavin dependent monooxygenases are classified into eight groups (A ~ H) based on their folding pattern and function [2]. Group D flavin dependent monooxygenases adopt an acyl-CoA dehydrogenase (ACAD) fold and depend on separate flavin reductases for the supply of reduced flavin (FAD or FMN) [4]. Most common reactions carried out by this class of monooxygenases are the electrophilic aromatic substitution as in 4-hydroxyphenylacetate 3-monooxygenase [6]. However, heteroatom oxidation reactions have also been reported in this class of enzymes, for example, sulfoxidase (dibenzothiophene monooxygenase [7]), N-hydroxylase (isobutylamine N-hydroxylase [8]) and nitrososynthase (ORF36 [9], and KijD3 [10] and DnmZ [11]) have been studied.

Here, we determined the Apo (PDB ID: 5GJ7) and FAD bound (5Z8T) structures of an *Alicyclobacillus acidocaldarius* protein which has been previously annotated as an Acyl-CoA dehydrogenase. *A. acidocaldarius* is acidophilic and thermophilic organism, growing at temperatures between 20 and 70 °C and pH ranges of 2 to 6. Due to its thermoacidic features, *A. acidocaldarius* serves as a model for biochemical studies of its enzymes. Although the overall structure is very similar to other Acyl-CoA dehydrogenases (ACADs), phylogenetic analysis of this protein showed that it is more closely related to other bacterial group D flavin dependent monooxygenases, which also adopt ACAD fold. Moreover, multiple sequence alignment with other Acyl-CoA dehydrogenases revealed that this protein lacks a conserved glutamate residue that is critical for ACAD's enzymatic activity by acting

as a base that abstracts a proton from acyl-CoA. Structural comparison of substrate binding pocket from ACAD and flavin dependent monooxygenases also indicated that the potential substrate binding pocket of 5Z8T share more similarity with flavin dependent monooxygenases. These findings suggest that 5Z8T is a bacterial group D flavin dependent monooxygenase.

## 2. Materials and Methods

### 2.1. Cloning and Protein Preparation

The putative Acyl-CoA dehydrogenase gene was amplified from *A. acidocaldarious* genomic DNA by polymerase chain reaction (PCR) using primers (5'TACTTCCAATCCAATGCAATGTACGACATCTA TGGAGAAG-3' and 5'-TTATCCACTTCCAATGTTATTACCTCCCCGCCTGCCCC-3').

The purified PCR product was cloned into pLIC-Tr3Ta-HA vector with an N-terminal His6-tag and TEV (Tobacco Etch Virus) protease cut site. After TEV protease treatment, three amino acids (SNA) from the vector sequence were left on the N-terminus of putative Acyl-CoA dehydrogenase protein. The construct was then transformed into BL21 (DE3) *E. coli* strain (Novagen). Cells were grown in LB medium containing 50 µg/mL carbenicillin at 37 °C until the OD600 reached 0.8~0.9 and isopropyl β-D-1-thiogalactopyranoside (IPTG) was added to a final concentration of 1 mM. Growth was continued for overnight at 24 °C, cells were harvested by centrifugation and lysed by sonication in lysis buffer composed of 20 mM Tris-HCl pH 7.5, 250 mM NaCl. The lysate was cleared by centrifugation, after which the supernatant was loaded onto a Ni-Sepharose 6 affinity column and eluted with a stepwise gradient of 50, 100, 200 and 400 mM imidazole in lysis buffer (20 mM Tris-HCl pH 7.5 and 250 mM NaCl). After the N-terminal $His_6$-tag was cut by TEV protease at 4 °C for 16 h, putative Acyl-CoA dehydrogenase was purified using a Superdex75 size-exclusion column (GE Healthcare) equilibrated with buffer composed of 20 mM Tris-HCl pH 7.5 and 250 mM NaCl, 2 mM dithiothreitol, and 2 mM EDTA. Purity of the protein was analyzed by 12% sodium dodecyl sulfate-polyacrylamide gel electrophoresis.

### 2.2. Crystallization, Data Collection, and Structure Determination

Purified protein was concentrated to 15.0 mg/mL by centrifugal ultrafiltration (Amicon) in 20 mM Tris-HCl pH 7.5 and 250 mM NaCl, 2 mM dithiothreitol, and 2 mM EDTA. The protein concentration was measured by Bradford assay. Crystals were obtained by the sitting-drop vapor-diffusion method at 20 °C by mixing 1 µL protein solution with 1 µL well solution. The well solution for apo-crystal contained 0.2 M Magnesium Chloride, 0.1 M MES: NaOH pH 6.5, 10% (*w/v*) polyethylene glycol (PEG) 4000. Flavin dependent monooxygenase: FAD complex crystals were grown with 1.5% PEG 4000 and 0.1 M Sodium Acetate: HCl pH 4.6 after 30 minutes incubation with 2 mM FAD. Both the apo and FAD complex crystals appeared in about 4 weeks. For cryoprotection, apo crystals were transferred into 0.1M MES: NaOH pH 6.5, 10% (*w/v*) polyethylene glycol (PEG) 4000, and 25% glycerol, and complex crystals were with 2 mM FAD and 1.5% (*w/v*) polyethylene glycol(PEG) 4000, 0.1 M Sodium Acetate: HCl pH 4.6, and 35% glycerol followed by flash-freezing in liquid nitrogen.

X-ray diffraction data of the apo and complex crystals were collected at a 1.95 Å and 2.3 Å resolution at PAL beamline (Korea) 7A and 5C, respectively. Data were processed with HKL2000 (HKL research Inc., Charlottesville, VA, USA) [12], and initial model of apo structure was obtained using the Molrep program of CCP4 package [13] with Acyl-CoA dehydrogenase (*M. thermoresistibile*) structure (PDB ID: 3NF4) as a search model (sequence identity of 26%). The initial model of the complex was solved using the molecular replacement program with apo structure as a search model. The space group of apo and complex was $P2_12_12$ and $I4_122$. The model was refined with REFMAC [14], and manual model building was performed using the COOT program [15]. Ten residues in central region (134 to 143) in apo model and four residues in C-terminal region (395 to 398) in apo and complex model was not observed. The Ramachandran plot of both structures showed 100% of residues are in the most favored or favored region [16]. The data collection and refinement statistics are summarized in

Table 1. The coordinate and structure factors for apo and FAD complex have been deposited in the RCSB (Research Collaboratory for Structural Bioinformatics) protein Data Bank with accession code 5GJ7 and 5Z8T, respectively.

**Table 1.** Data collection and refinement statistics.

|  | **Apo (5GJ7)** | **FAD Bound (5Z8T)** |
|---|---|---|
| **Data Collection** | | |
| Space group | $P2_12_12$ | $I4_122$ |
| Unit Cell Dimensions (Å) | a = 155.7, b = 69.4, c = 83.3 | a = b = 177.3, c = 285.3 |
| Resolution (Å) [a] | 20-1.95 (1.98-1.95) | 20-2.35 (2.39-2.35) |
| Observed reflections | 1,136,579 | 4,407,376 |
| Unique reflections | 65,742 | 95,626 |
| Completeness (%) | 99.6 (100) | 100 (100) |
| $R_{sym}$ [b] | 0.10 (0.65) | 0.14 (0.66) |
| I/σ(I) [c] | 19.1 (4.7) | 19.6 (4.0) |
| **Refinement** | | |
| $R_{cryst}$ (%)/$R_{free}$ (%) [d] | 19.8/22.3 | 22.4/25.6 |
| No. of Residues | 776 | 1182 |
| No. of FAD | - | 3 |
| No. of Water molecules | 271 | 610 |
| R.m.s deviations | | |
| Bond lengths (Å) | 0.016 | 0.017 |
| Bond angles (°) | 1.667 | 1.605 |

[a.] Resolution range of the highest shell is listed in parentheses; [b.] $R_{sym} = \sum |I - \langle I \rangle| / \sum I$, where I is the intensity of an individual reflection and $\langle I \rangle$ is the average intensity over symmetry equivalents; [c.] I/σ (I) is the mean reflection intensity/estimated error; [d.] $R_{cryst} = \sum ||Fo| - |Fc|| / \sum |Fo|$, where Fo and Fc are the observed and calculated structure factor amplitudes, $R_{free}$ is equivalent to Rcryst but calculated for a randomly chosen set of reflections that were omitted from the refinement process.

## *2.3. Phylogenetic Analysis*

Protein sequences used in the phylogenetic analysis were aligned by Muscle method. The evolutionary history was inferred using the Maximum Likelihood method based on the JTT matrix-based model [17]. The bootstrap consensus tree inferred from 1000 replicates is taken to represent the evolutionary history of the taxa analyzed [18]. Branches corresponding to partitions reproduced in less than 50% bootstrap replicates are collapsed. The percentage of replicate trees in which the associated taxa clustered together in the bootstrap test (1000 replicates) are shown next to the branches [18]. The analysis involved 9 amino acid sequences and all positions containing gaps and missing data were eliminated. There was a total of 350 positions in the final dataset. Evolutionary analyses were conducted using MEGA7 [19].

## 3. Results and Discussion

### *3.1. Overall Structure*

The Apo and FAD bound complex structures of *A. acidocaldarius* protein previously annotated as an Acyl-CoA dehydrogenase were determined to 1.95 Å and 2.35 Å, respectively. The Apo and FAD bound complex structures are very similar with each other with RMSDs (root-mean-square deviations) of 1.0 Å when 332 Cα atoms out of 394 residues are compared by Pymol program and both structures formed a tetramer with dimer of dimers assembly (Figure 1). There are two kinds of dimer interfaces in the tetramer. The first interfaces (between magenta and yellow and cyan and green monomers) with a buried surface area of 1058 Å², provide four FAD binding sites in the interfaces. The other dimeric interaction (between magenta and cyan and yellow and green) has an interface area of 1314 Å² that is mainly contributed by C-terminal α helix (M375-G394) extending into other monomer, which is one of the characteristics of ACAD's fold proteins.

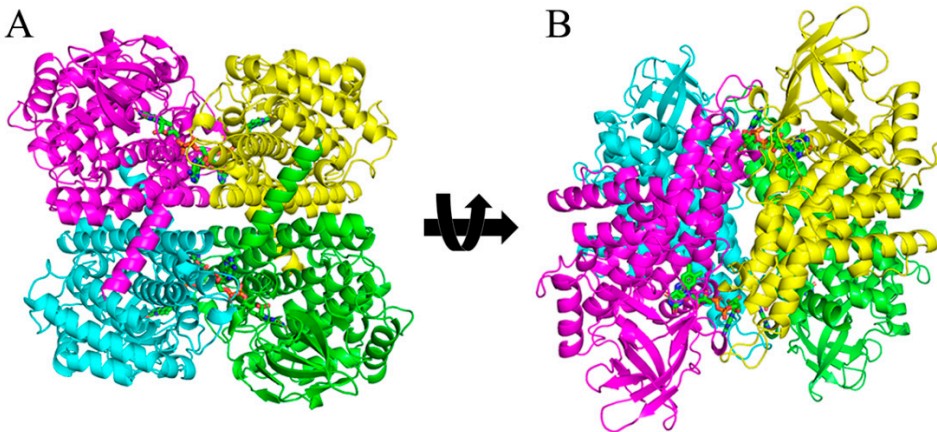

**Figure 1.** The overall structure of FAD bound complex. (**A**) Tetramer with dimer of dimers assembly of 5Z8T is drawn in cartoon representation. The dimers which provide FAD binding sites are colored by magenta–yellow and cyan–green. FAD molecules are shown in the ball-and-stick model. (**B**) The 90° rotated view of (A).

### 3.2. FAD and Potential Substrate Binding Site

The FAD binding site is formed in the first dimer interface mentioned above (interface between cyan and green monomers is shown in Figure 2A). Nonpolar residues such as L89, W93, F163, W201, M206, L363, and A368 form hydrophobic contacts with the dimethylbenzene part of the isoalloxazine ring of FAD. The part of isoalloxazine ring that contain carbonyl and amide group makes hydrogen bonds to N127, S128, A130, T131, and T165 (Figure 2A, dashed lines). The isoalloxazine ring of the FAD molecule is deeply buried in the protein, whereas the adenosine moiety of FAD points toward the solvent (Figure 2B). Comparison of the FAD binding sites of the apo and FAD complex structure showed that the overall B-factors around FAD binding site was decreased from 38.4 (apo) to 23.4 (FAD complex). Also, disordered loop region (residues 134–143) in the apo structure was visible in the FAD complex structure. One of the features of 5Z8T structure is the presence of a pocket close to the isoalloxazine ring of bound FAD. The width and depth of the pocket are approximately 30 Å, and 20 Å, respectively, and the volume is about 310 Å$^3$ calculated by the KVFinder program [20,21]. Comparison of substrate binding sites of our structure and structural homologues found by Dali (1JQI and 4KCF, top scoring ACAD and flavin-dependent monooxygenase, respectively) showed that the location of substrate binding pocket is commonly next to the isoalloxazine ring of bound FAD or FMN. However, the shape of the putative substrate binding pocket of 5Z8T is distinct from Acyl-CoA dehydrogenases and similar to that of flavin dependent monooxygenase such as *Actinomadura kijaniata* Kijd3 [10]. *A. kijaniata* Kijd3 uses FMN to oxidize dTDP-3-amino-2,3,6-trideoxy-4-keto-3-methyl-D-glucose (dTDP-sugar) and contains shallow and wide substrate binding pocket (Figure 2B). The shape of this pocket is distinct from those observed in ACADs such as 1JQI (rat short chain acyl-CoA dehydrogenase) which contain long and narrow pocket that can accommodate the long hydrocarbon tail part of Acyl-CoA (Figure 2C). From these observations, the pocket close to the isoalloxazine ring is likely to be a potential substrate binding site of 5Z8T and is similar to other flavin-dependent monooxygense in its shape.

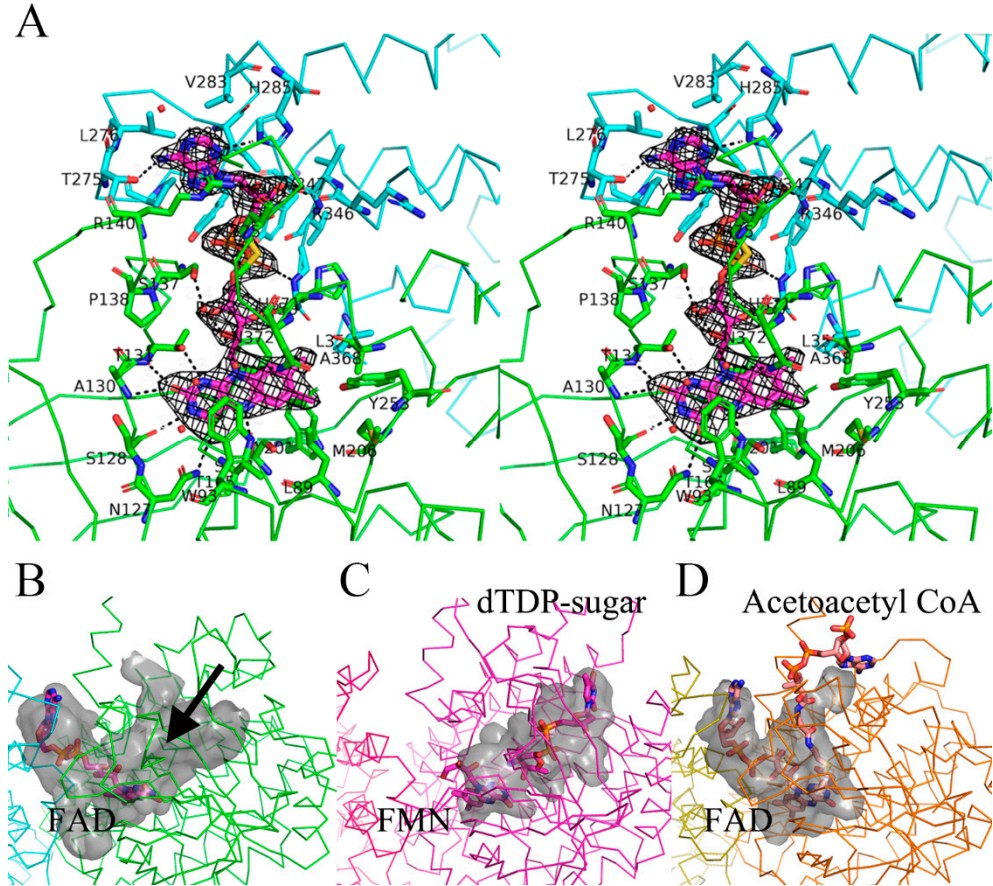

**Figure 2.** Analysis of FAD and potential substrate binding site. (**A**) Stereoview of FAD binding site. Residues interacting with FAD are shown in the ball-and-stick model. Hydrogen bonds are represented by dashed lines. Water molecules are indicated by red spheres. Final $2F_o$-$F_c$ electron density map of FAD contoured at 1.5 σ is shown. (**B**) Close up view of putative substrate binding site drawn by surface cavity mode in Pymol. The FAD molecule of 5Z8T are drawn in the ball-and-stick model and the potential substrate binding pocket is indicated by arrow. (**C**) Substrate binding site of 4KCF (flavin dependent monooxygenase) with bound FMN (Flavin mononucleotide) and dTDP-3-amino-2,3,6-trideoxy-4-keto-3-methyl-D-glucose (dTDP-sugar)-sugar (**D**) Substrate binding site of 1JQI (acyl-CoA dehydrogenase) with bound FAD and acetoacetyl-CoA.

## 3.3. Features of Group D Flavin Dependent Monooxygenase

To identify the function of 5Z8T, structural homologs of 5Z8T were searched using the Dali server [22] (Supplemental Table S1). The top 20 scorers include Acyl-CoA dehydrogenases (ACADs) and group D flavin dependent monooxygenases. The most similar protein is short chain Acyl-CoA dehydrogenase (PDB code 1JQI, Z score/RMSD value 44.4/2.3) from *Rattus norvegicus* [23]. The other similar proteins are Medium Chain Acyl-CoA dehydrogenase (3MDE, 44.1/ 2.1) from *Sus scrofa* [24] and short/branched chain specific Acyl-CoA dehydrogenase (2JIF, 44.0/ 2.0) from *homo sapiens* (Supplementary Figure S1). Similar to our structure 5Z8T, all of the ACAD homologs found by a Dali search formed a tetramer and bind FAD molecule as a cofactor. However, sequence alignment with other ACAD's revealed that a glutamate, a key residue that is highly conserved as a catalytic residue in most Acyl-CoA dehydrogenase family [25] is absent in 5Z8T (Supplemental Figure S2).

Structurally similar proteins other than ACAD family include Kijd3 (PDB ID: 4KCF, Z score/RMSD value 43.6/ 2.4) [10], DszC (3X0Y, 43.2/2.1) [7], ORF36 (3MXL, 42.7/2.5) [9], and DnmZ (4ZXV, 42.6/2.3) [11], all of which are group D flavin dependent monooxygenases. In contrast to the ACAD family, these proteins do not require a catalytic glutamate residue in the active site. To further investigate the

relationship with these structural homologs, we carried out phylogenetic analysis. We compiled a set of 10 bacterial homologs of 5Z8T in the top 20 scorers of the Dali search, and used the Muscle and Maximum likelihood method [26] to construct a phylogenetic tree using the program MEGA [19] (Figure 3). Phylogenetic analysis showed that 5Z8T belonged to the same clade as other flavin dependent monooxygenases, such as DszC, DnmZ, and Kijd3, and was distant from ACAD's. This phylogenetic analysis suggests that the function of 5Z8T is related to group D flavin dependent monooxygenase and different from ACAD's.

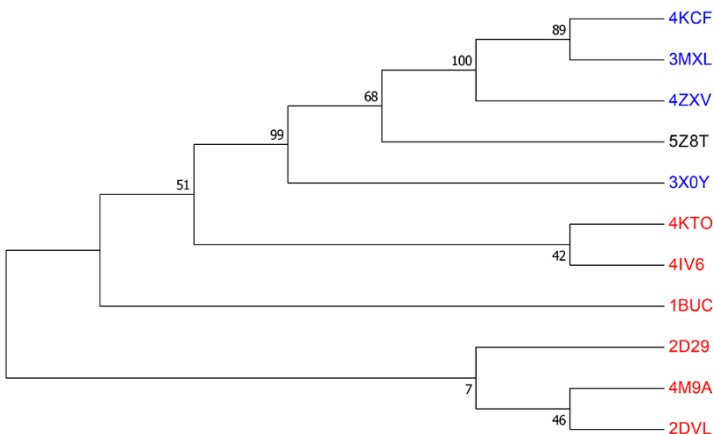

**Figure 3.** Phylogenetic analysis of 5Z8T with bacterial homologs. The phylogenetic tree of 11 taxa including 5Z8T, flavin dependent monooxygenases and Acyl-CoA dehydrogenases. The group D flavin dependent monooxygenases are shown in blue, the Acyl-CoA dehydrogenases in red. (For further detail, see the Supplementary Table S1 and Supplementary Figure S1).

In summary, the Apo (PDB ID: 5GJ7) and FAD bound complex (5Z8T) structures of *A. acidocaldarius* protein previously annotated as an Acyl-CoA dehydrogenase showed the typical characteristics of ACAD fold protein—tetramer formation with extended C-terminal helix and FAD binding sites in the dimer interfaces. However, 5Z8T lacks the conserved catalytic glutamate residue, which is a common feature of Acyl-CoA dehydrogenases and phylogenetic analysis revealed that 5Z8T was grouped with other flavin dependent monooxygenases rather than ACADs. Furthermore, examination of the potential substrate binding pocket of 5Z8T showed that its shape is different from the long and narrow binding pocket of ACADs and more similar to the substrate binding pocket of *A. kijaniata* Kijd3, a flavin dependent monooxygenase. These findings suggest that 5Z8T is a novel bacterial group D flavin dependent monooxygenase. The exact nature of its substrate and function need further characterization. We believe our study illustrates assigning a protein's function only by overall sequence identity can be misleading especially the sequence identity is not very high.

**Supplementary Materials:** The following are available online at http://www.mdpi.com/2073-4352/9/11/548/s1, Figure S1: superposition of 5Z8T with structural homologs found by the Dali server. Structural homologs of 5Z8T were searched using the Dali server and the top three scoring hits are superposed with 5Z8T. 5Z8T is colored green and structural homologs 1JQI (*Rattus norvegicus* short Chain Acyl-CoA Dehydrogenase), 3MDE (*Sus scrofa* medium chain Acyl-CoA dehydrogenase), and 2JIF (*Homo sapiens* short/branched chain Acyl-CoA dehydrogenase) are colored blue, yellow, and orange, respectively. Bound substrate molecules (FAD and acyl CoA) are drawn in ball-and-stick model. Figure S2: Multiple sequence alignment of 5Z8T and 10 bacterial homologs identified by the Dali search. Multiple sequence alignment of 10 bacterial homologs of 5Z8T in the top 20 scorers of the Dali search. A glutamate residue that is highly conserved as a catalytic residue in most Acyl-CoA dehydrogenase family is shown in red box and highlighted with red asterisk. 4KTO, a *Sinorhizobium meliloti* isovaleryl-CoA dehydrogenase contains catalytic glutamate at different position (E254). This key glutamate residue is absent in group D flavin dependent monooxygenases including 5Z8T (from *Alicyclobacillus acidocaldarius* species), 3X0Y (*Rhodococcus erythropolis*), 4ZXV (*Streptomyces peucetius*), 3MXL (*Micromonospora carbonacea*), and 4KCF (*Actinomadura kijaniata*). Table S1. Summary of structural homologs of 5Z8T found by a Dali search. Results of the Dali search for structural homologs of 5Z8T. Top 20 scorers are ranked by Z-score and their PDB ID, RMSD with 5Z8T, sequence identity, species name, and description are listed.

**Author Contributions:** H.M., S.S., and J.C. designed the experiments; H.M. determined the apo structure and S.S. determined the FAD bound structure.

**Funding:** This work was supported by NRF-2018R1D1A1A09083579 grant and by the 2017 sabbatical year research grant of the University of Seoul.

**Acknowledgments:** The authors thank the staff members of Pohang Synchrotron Radiation beamlines 5C for assistance in data collection and HB Oh and HY Jang at the Sogang University for mass spectrometry analysis.

**Conflicts of Interest:** The authors declare no conflict of interest.

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
