# Peer review of "Crystal Structures of Putative Flavin Dependent Monooxygenase from Alicyclobacillus Acidocaldarius"

_crystals, doi:10.3390/cryst9110548_

Round 1
Reviewer 1 Report
This short article reports the crystals structure ot the apo and FAD bound form of a flavin dependent monooxygenase.
THe work is sound but provides little insight into the enzyme function (other than broad classification). It appears from the text that other examples already exist of enzymes with a similar fold that are flavin monooxygenases as opposed to Acyl-CoA dehydrogenases. It would have been a much more interesting article if the authors had tried to establish flavin moonoxygenase activity with any substrates (even if it isn't the natural substrate).
They could also provide more info on the crystal structure - show electron density of FAD. Show overlap of active site and FAD binding site with other flavin monooxygenases and Acyl-CoA dehydrogenases to highlight similarities and differences.
Line 10 add the A-H classification to the introduction
Line16 give examples of flavin monooxygenases it is related to.
Line 40 why is this glutamate critical for dehydrogenase activity
Line 64 70 and 71 Sometimes the authors put a space between the number and units sometimes they don't. The need to consistently add a space (eg line 64; 2 mM EDTA).
Line 137 The pocket
Figure S1 and S2 Provide the details of the enzymes and organisms in the figure legend for completeness
Reviewer 2 Report
This manuscript describes the structure determination and analysis of the Alicyclobacillus acidocaldarius protein formerly annotated as an Acyl-CoA dehydrogenase (ACAD) in apo and FAD bound state at 1.95 and 2.35 Å, respectively. The apo form structure was determined by molecular replacement using 3NF4, an Acyl-CoA dehydrogenase from M. thermoresistibile as a template and the holo form structure was solved using molecular replacement using the apo structure as a search model.
Overall the crystallography and analyses seem to have been done reasonably well, but the overall presentation could be improved. A few specific points that the authors should consider for improving the manuscript are specified bellow.
The authors hypothesized based on structural similarity and substrate binding pocket comparison that the enzyme is more closely related to other bacterial group D flavin-dependent monooxygenases. The hypothesis put forward is interesting, but it should be further supported. The work presented is very scarce, only the structures in the presence and absence FAD were obtained but no biochemical investigation was performed. Also, the motivation of obtaining these structures is not clear. The authors should include what it is the importance to understand this protein.
Comments:
Line 58: In which temperature was it kept overnight?Line 59: Lysis buffer composition (DNAse, RNAse, protease inhibitor?)
Line 60: Include the description of the stepwise gradient and the elution buffer
Line 67: What method was used to quantify protein sample?
Line 68: The enzyme stock buffer is not described
Line 69: Drop size and ratio of reservoir and protein volume were not mentioned. How long did it take to grow the protein crystals?
Line 71: Was the enzyme pre incubated with FAD? For how long? 200 mM FAD seems to be too high; it must be a typo. Double check the experimental conditions.
Line 79: What is the percentage of identity of the model (3NF4) used as a template in the molecular replacement compared to the enzyme?
Which software of the CCP4 package was used to perform the molecular replacement?
The comparison between apo and holo form of the enzyme could be extended to parameters other than just the global RMSD. It would be interesting to check the B factor for the regions around the flavin to analyze how the presence of the flavin changes the protein dynamics.
Figure 1. FAD molecules are not clear
The labels in figure 2A should be more clear.
Is there any way to gain insight on the actual physiological role of this enzyme based on the phylogenetic analysis showed in the paper?
Reviewer 3 Report
The manuscript submitted by Dr. Jungwoo Choe describes the crystal structures of the apo and FAD bound forms of a protein from A. acidocaldarius, which was previously considered to be an Acetyl-CoA dehydrogenase and which is more likely to be a monooxigenase, based on the data presented in the manuscript.
The topic is certainly interesting and the manuscript is clear.
There are three main observations to be made.
(a) Is it possible to delineate an experimental validation, at least in vitro, of these hypotheses?
(b) The Authors might consider to comment the fact that A. acidocaldarius is quite an exotic organism.
(b) Is there in the genome of this organism a Glu-containing protein that might be an Acetyl-CoA dehydrogenase with a 3D structure close to those presented in the manuscript? A homology model would be sufficient for the moment.
Minor considerations.
Lines 114-115. The Apo and FAD bound structures are very similar (RMSD = 1 Angstrom). I have four observations.
(i) I suppose that the RMSD was computed by using only the Calpha atoms. However, this should be written in the text.
(ii) Moreover, it is necessary to indicate the percentage of residues used to compute the RMSD (usually, some of them are too divergent and are not used to minimize the RMSD).
(iii) It is necessary to indicate the program (or the algorithm used to minimize the RMSD).
(iv) Might the Authors spend some time to comment why the space groups are different (P21212 and I4122) despite the structures are nearly identical?
Line 115. Might the Authors explain the mining of the expression (dimer of dimers). Is it known if the protein is tetrameric in physiological conditions (in vivo)?
Line 116. Is the FAD binding site solvent accessible or it is completely buried at the interface? Analogously, is the substrate binding site solvent accessible in the crystal (solid) state?
Line 134. “... is approximately...” should be “ are approximately ...”
Line 142. “... was superposed with 5Z8T ...” should be “was superposed to 5Z8T...” Moreover, one is expecting to see the RMSD and related details here.
Round 2
Reviewer 3 Report
The revised version of the manuscript is significantly better than previous version. The Authors did a considerable effort to answer all my questions.